# Gaze Estimation Based on Convolutional Structure and Sliding Window-Based Attention Mechanism

**DOI:** 10.3390/s23136226

**Published:** 2023-07-07

**Authors:** Yujie Li, Jiahui Chen, Jiaxin Ma, Xiwen Wang, Wei Zhang

**Affiliations:** 1School of Artificial Intelligence, Guilin University of Electronic Technology, Guilin 541004, China; yujieli@guet.edu.cn (Y.L.); 2001630104@mails.guet.edu.cn (J.C.); 2001630112@mails.guet.edu.cn (J.M.); 2001630202@mails.guet.edu.cn (X.W.); 2Guangxi Colleges and Universities Key Laboratory of AI Algorithm Engineering, Guilin 541004, China

**Keywords:** gaze estimation, swin transformer, convolutional neural networks (CNN), deep learning, self-attention mechanism

## Abstract

The direction of human gaze is an important indicator of human behavior, reflecting the level of attention and cognitive state towards various visual stimuli in the environment. Convolutional neural networks have achieved good performance in gaze estimation tasks, but their global modeling capability is limited, making it difficult to further improve prediction performance. In recent years, transformer models have been introduced for gaze estimation and have achieved state-of-the-art performance. However, their slicing-and-mapping mechanism for processing local image patches can compromise local spatial information. Moreover, the single down-sampling rate and fixed-size tokens are not suitable for multiscale feature learning in gaze estimation tasks. To overcome these limitations, this study introduces a Swin Transformer for gaze estimation and designs two network architectures: a pure Swin Transformer gaze estimation model (SwinT-GE) and a hybrid gaze estimation model that combines convolutional structures with SwinT-GE (Res-Swin-GE). SwinT-GE uses the tiny version of the Swin Transformer for gaze estimation. Res-Swin-GE replaces the slicing-and-mapping mechanism of SwinT-GE with convolutional structures. Experimental results demonstrate that Res-Swin-GE significantly outperforms SwinT-GE, exhibiting strong competitiveness on the MpiiFaceGaze dataset and achieving a 7.5% performance improvement over existing state-of-the-art methods on the Eyediap dataset.

## 1. Introduction

Gazing is a crucial form of human behavioral information that contains a wealth of psychological insights and is a vital clue for comprehending human intentions and emotions [1]. Gaze estimation by the human eye has a wide range of applications, including medical treatment [2,3], virtual reality [4], human–computer interaction [5,6,7,8], market research [9], and other fields. Gaze estimation research can be broadly categorized into three areas based on different scenarios and applications: gaze point prediction [10], gaze target prediction [11], and three-dimensional gaze estimation [12,13]. This study focuses on three-dimensional line-of-gaze estimation.

Gaze estimation methods can be roughly divided into two categories: model-based and appearance-based. The model-based gaze estimation method typically involves using eye information, such as iris radius, kappa angle, and pupil position, to create a geometric model for prediction [14,15]. These methods frequently require specialized equipment to capture specific eye information [16,17], which can be costly and have limited applications. As shown in Figure 1a, the appearance-based gaze estimation method does not require specialized equipment and directly learns mapping functions from images in the gaze direction, but requires enormous training data. Traditional appearance-based gaze estimation methods [18,19,20] typically learn theme-specific mapping functions. Therefore, these methods only demonstrate satisfactory performance in scenarios with limited head posture and subject constraints and perform poorly in unconstrained scenarios. As shown in Figure 1b, Sugano et al. [20] utilized a technique that integrated a saliency map, and generated a gaze probability map and the average eye image to optimize the gaze estimator. This approach enables the accurate prediction of gaze points using an optimized gaze estimator. With the emergence of deep learning and the availability of numerous datasets, researchers [21,22,23,24,25,26,27,28] have proposed appearance-based deep-learning gaze-estimation methods. As shown in Figure 1c, these methods utilize various convolutional neural networks (CNN) models and demonstrate exceptional performance even in uncontrolled environments, thereby enhancing the accuracy of gaze estimation prediction in such scenarios [28]. Researchers have achieved good performances using various CNN models. CNNs have a strong ability to extract spatial details and effectively capture the local features of images. However, CNNs may lose significant amounts of valuable information during pooling. This loss of information can hinder the identification of local and global correlations, making it difficult to capture the interrelationships between the eyes and face.

Transformer was first proposed by Vaswani et al. [29]. At this stage, the transformer model achieved the most advanced performance in natural language processing tasks. With the remarkable performance of transformers in various tasks and the development of models based on visual attention mechanisms in recent years, an increasing number of transformer-based models have been applied to computer vision tasks and have achieved good performance. Vit [30] employed a transformer structure to obtain better results than the most advanced CNN in image classification tasks, showing that the transformer achieved a better global relationship capture ability than the aforementioned CNN. However, unlike fixed-length word vectors in natural language processing tasks, target feature scales in visual tasks are different, and fixed-length word vectors in transformers are unsuitable for multiscale feature learning. Simultaneously, the resolution of the image is much higher than the number of words in the text paragraph. therefore, it is difficult to achieve the global self-attention calculation. To solve these problems, Liu et al. [31] proposed a Swin Transformer model in 2021. Compared to transformer, the Swin Transformer can better learn multiscale features by building a hierarchical feature map through a multistage structure and employs Windows Multi-head Self-Attention (W-MSA) to limit self-attention computing within the window, significantly reducing the computational complexity of self-attention, while designing Shifted Windows Multi-head Self-Attention (SW-MSA) to ensure its global modeling ability.

Although numerous studies demonstrate the impressive capabilities of utilizing transformers in visual tasks, their performance still lags behind that of similarly sized CNN. One limiting factor is that other transformer-based models, such as the vision transformer (ViT) [30], segment input images into non-overlapping patches [32] using a slicing-and-mapping mechanism. The Swin Transformer follows the ViT approach [30] by dividing the image into non-overlapping patches. These patches are then converted into vectors and inputted into subsequent structures. As a result, the two-dimensional structure and local spatial information inside the patch are lost, limiting the ability of the network to learn the local spatial features of the image.

In this paper, we developed a Swin Transformer-based model to address gaze estimation tasks. Based on the tiny version of the Swin Transformer, Swin-T, we designed two network models, SwinT-GE and Res-Swin-GE. SwinT-GE uses Swin-T as the primary framework to perform direct gaze estimation on images. To address the problem of the slicing-and-mapping mechanism damaging the patch spatial information, ResSwin-GE utilizes a combination of ResNet18 [33] and Swin-T to perform feature mapping on the image. Compared with SwinT-GE, this approach preserves more local spatial features while leveraging the Swin-T structure for multiscale feature learning and global modeling of feature mapping.

To the best of our knowledge, this is the first study to introduce the Swin-T model to a gaze-estimation task. The main contributions of this study are as follows.

(1)We introduced Swin-T into gaze estimation tasks and proposed a gaze estimation network model based on Swin-T.(2)We proposed a novel network model that combined a CNN and Swin-T and achieved improved performance on two public datasets.(3)We analyzed the improvement in network performance caused by using a CNN as a pre-feature extraction module.

The rest of this paper is organized as follows. In Section 2, we review some related work. Section 3 introduces the model structure in detail. In Section 4, we present our experimental results and analyze the performance of the model. Finally, Section 5 summarizes the study and suggests future research plans.

## 2. Related Works

In recent years, appearance-based deep-learning methods have become popular in the field of gaze estimation, and numerous CNN-based gaze estimation methods have been proposed. Zhang et al. [21] proposed the first CNN-based gaze-estimation method. They designed a simple CNN based on LeNet [34], which employs a single grayscale image of the human eye as an input for predicting the gaze direction. The performance of the network exceeds that of most traditional appearance-based gaze-estimation methods. Several researchers have proposed gaze-estimation methods based on CNNs. Fischer et al. [22] utilized two VGG16 networks [24] to extract individual features from two human eye images that were then connected for regression analysis. Cheng et al. [24] developed a four-stream CNN for extracting features from two human eye images. Two streams were utilized to extract individual features from the left and right eye images, whereas the other two streams were used to extract joint features from both images. Cheng et al. [25] employed asymmetric regression to address extreme head postures and lighting conditions. Park et al. [26] developed a method for learning the representation of human eye images that reduced the impact of individual appearance differences. Chen et al. [27] utilized extended convolutional networks to detect subtle changes in eye images. CNN-based methods have made significant progress in gaze estimation. However, CNN-based methods commonly suffer from information loss in the pooling layer. The pooling layer compresses the information within the input feature map into smaller representations. However, this process may result in the loss of certain spatial information. The loss of information becomes more severe as the number of pooling layers increases, which hinders global network modeling. Therefore, the proposed method reduces the number of pooling layers and introduces a self-attention mechanism for global modeling. The self-attention mechanism calculates the dependency relationship between each feature vector and all other feature vectors. It assigns a weight to each feature vector to demonstrate its importance within the global features, thereby conducting global modeling. Compared to traditional convolutional layers, the self-attention mechanism can model global features more flexibly without requiring the pooling of input feature maps, thus avoiding information loss.

Dosovitskiy et al. [30] were the first to propose the application of transformers in the field of computer vision and achieved impressive results. In the image classification task, ViT uses the pure transformer model to achieve better performance than the most advanced convolutional network. Compared with CNN, the transformer model better captures global relationships. Inspired by this, Cheng et al. [35] proposed the application of a transformer model to gaze estimation. However, the slicing-and-mapping mechanism used in this method destroys spatial information contained in the patch. Consequently, the pure transformer model fails to achieve good results in the field of gaze estimation. Hence, they proposed a hybrid CNN and Transformer model for gaze estimation, which preserves the local spatial information of facial images and enhances the performance of the model. The structure of traditional transformers is unsuitable for learning multiscale features in gaze estimation tasks, such as fine-grained human eye features that focus on local details and coarse-grained facial features that focus on overall quality. In addition, the global self-attention calculation for images is highly complex. Regarding the issue of the slicing-and-mapping mechanism destroying patch spatial information, this study proposes ResNet18 [33] in its stead to address the problem. ResNet18 can better preserve the spatial relationships between pixels within a patch, thereby maintaining the spatial information of the image. Residual connections enable the network to learn complex image features while maintaining information flow. Considering that traditional transformers are unsuitable for gaze estimation tasks, this study adopted Swin-T, which is suitable for computer vision tasks and has good computational efficiency for self-attention calculations and multiscale feature learning.

## 3. Gaze Estimation Based on Swin-T

In this section, we introduce the experimental details of the application of Swin-T to gaze estimation. Two structures were designed for gaze estimation, SwinT-GE and Res-Swin-GE.

### 3.1. SwinT-GE Applied to Gaze Estimation

SwinT-GE uses the tiny version Swin-T infrastructure in the Swin Transformer to conduct experiments. The network structure of SwinT-GE is shown in Figure 2, and its main parts are as follows.

Input a face image $I\in R^{H\times W\times 3}$. Patch partition: the face image is divided into $\frac{H\times W}{16}$ non-overlapping patches according to adjacent 4×4 pixels and each patch $I_{i}\in R^{H\times W\times 3}$, where $i=1\cdots \frac{H\times W}{16}$. Then, each patch is flatten according to the three channels of R, G, B to obtain the feature map $I^{\prime }\in R^{\frac{H}{4}\times \frac{W}{4}\times 48}$, where *H* and *W* refer to the height and width of the image, respectively.

The feature map, which are the stage-wise outputs of different layers, undergoes a multi-stage process to construct feature maps of varying sizes, enabling multiscale self-attention calculations. Stage 1: the number of channels in the feature map is linearly mapped from 48 to *C* using Linear Embedding (where C=96 in Swin-T). The remaining three stages (Stage 2–4) are first downsampled through patch merging, which halves the height and width of the feature map, doubles the number of channels, and then enters the repeatedly stacked Swin-T Block for self-attention calculation. The core module in each stage is the Swin-T Block. After the feature map enters the Swin-T Block, it is first normalized through the Layer Norm [36] to stabilize the data distribution, followed by W-MSA or SW-MSA. Compared with MSA in ViT, W-MSA significantly reduces the number of calculations. MSA needs to perform self-attention calculations on the entire feature map. Whereas, W-MSA first divides the feature map into multiple mutually independent overlap windows of equal length and width and then performs self-attention calculations on the feature vectors inside each window, effectively reducing the amount of calculation. However, only a self-attention calculation is performed on the feature vector in a single window, and the information between windows cannot be transmitted, which greatly limits the learning ability of the model and its global modeling ability. To address this challenge, Swin-T introduces SW-MSA, a mechanism that modifies the original window layout of W-MSA. SW-MSA achieves this by shifting the window layout diagonally from the upper left to the lower right, with a movement of half the window length. Consequently, the feature map is partitioned into new windows based on different eigenvectors of the modified window layout. Subsequently, the self-attention is calculated for the eigenvectors in the new window to achieve information transmission between different windows and ensure the global modeling ability of the model. Therefore, W-MSA and SW-MSA always appear in pairs. A DropPath layer is used to regularize the feature map obtained from the self-attention calculation. Then, the regularized feature map is added to the feature map before entering the Layer Norm to form a residual connection, the next Layer Norm for data normalization, and a multilayer perceptron (MLP) to increase the nonlinear expression ability of the model. Adding the feature map from the DropPath layer to the feature map can form a residual connection output of the Swin-T Block before the feature map enters the Layer Norm.

To adapt to the gaze estimation task, this study adds Layer Norm, an adaptive average pool layer, and a fully connected layer with an output dimension of two to predict the gaze vector after the last stage.

### 3.2. Res-Swin-GE Applied to Gaze Estimation

Res-Swin-GE consists of a ResNet18-based ResNet Block and SwinT-GE with removed patch partitions and linear embedding modules. The ResNet was proposed by He et al. [33]. The ResNet network incorporates a residual structure by introducing residual connections between different layers. This design allows for the retention of shallow network features and effectively mitigates the issue of model degradation. As a result, ResNet enables the construction of deep convolutional neural networks that exhibit improved performance.

The overall network structure of Res-Swin-GE is shown in Figure 3. This study replaces the patch partition and linear embedding parts in SwinT-GE with ResNet Block, whose main structure is the basic structure ResNet18. To match the number of feature channels of the Swin-T Block in Stage 1, a 1 × 1 convolutional layer is used to adjust the number of output channels at the end of the ResNet Block. Input a face image $I\in R^{224\times 224\times 3}$, and obtain the feature map $I^{\prime }\in R^{7\times 7\times 96}$ after convolution partial feature extraction. Notably, this study does not adjust the height and width of the feature map, but directly inputs the feature map into the Swin-T Block. To match the size of the feature map, this study adjusts the window size in Swin-T to enhance accommodation. The self-attention calculation under small feature maps is discussed in detail in the subsequent experimental Section 4.4. The subsequent structure is consistent with that of SwinT-GE described in Section 3.1. After Stage 4, Layer Norm, adaptive average pooling layer, and a fully connected layer with an output dimension of two are added to predict the gaze vector.

## 4. Experiments

This section details the experimental performance of SwinT-GE and Res-Swin-GE on two public datasets, MpiiFaceGaze and Eyediap, and compares the effectiveness of the two primary modules in Res-Swin-GE. In Section 4.1, the experimental equipment and details are introduced. Section 4.2 provides a detailed description of the two public datasets used in this study, MpiiFaceGaze and Eyediap, as well as the preprocessing operations applied to the datasets. In Section 4.3, the evaluation metrics and methods for the model performance are described in detail. Section 4.4 discusses the effect of the hyperparameter Window Size on the model performance in Res-Swin-GE. In Section 4.5, the experimental results are analyzed, and some current advanced gaze estimation methods are selected and compared with the proposed method. In Section 4.6, Res-Swin-GE is subjected to ablation experiments to explore the impact of the different components of the model on the performance of the model.

### 4.1. Experimental Details

The model training in this study was conducted via a remote connection to a server with an Ubuntu 20.04.3 operating system and PyTorch deep learning framework, using Python3.8 programming language, and two NVIDIA GeForce RTX 3090 GPUs with 24 GB memory. ADAMW [37] was used as the optimizer, and L1Loss was used as the loss function for training. The iteration cycle was set to 30, and the initial learning rate was set to 5 × 10^−4^.

### 4.2. Datasets

In order to better evaluate the performance of the model, this article conducted experiments on two publicly available popular datasets, MpiiFaceGaze and Eyediap. Figure 4 shows examples of facial images from the two datasets.

The MpiiFaceGaze [38] dataset, proposed and publicly released by Zhang et al. in 2017, is a popular appearance-based gaze estimation dataset. The participants were requested to install the corresponding program on their laptops and gaze at the target points, which were generated at 20 random positions by the program every 10 min. The program would simultaneously call the front-facing camera of the computer to capture the facial data of the participants while generating the target points. The dataset comprised 213,659 facial images from 15 participants. This data collection method in real environments provides rich lighting and head-pose variations, offers unprecedented authenticity, and is suitable for evaluating unconstrained gaze estimation methods.

The Eyediap [39] dataset was proposed and publicly released by Kenneth et al. in 2014. Participants were requested to sit in front of a depth camera and continuously gaze at a randomly moving ball, while data were acquired using both the depth camera and a regular RGB camera. After data acquisition, the researchers used the three-dimensional coordinates recorded by the depth camera to calculate the gaze direction. The dataset contains 94 video segments from 16 participants, each accompanied by a file providing important information, such as the head pose and target location for each frame. The gaze-labeling results of the dataset were relatively accurate because of the use of a depth camera for data acquisition. However, there was no significant variation in lighting conditions because the data were collected exclusively in a laboratory environment.

This study followed the dataset preprocessing procedure proposed by Fischer et al. [22] to crop 224 × 224 color facial images from the two datasets for gaze estimation.

### 4.3. Evaluation Metric

In this study, the leave-one-person-out (LOPO) method was used as the experimental evaluation metric. Such an experimental strategy is common in experimental evaluations of gaze estimation. There are 15 experimental subjects in the MpiiFaceGaze dataset. One of the experimental subjects was selected successively as the test set, and the remaining 14 experimental subjects were selected as the training set for the experiments. Finally, the average angular error of all experiments was used as the model performance. This study adopted the same strategy for the Eyediap dataset. Notably, the Eyediap dataset lacks experimental subjects P12 and P13, with only 14 subjects.

In this study, angular error was used as the evaluation metric for model performance. The larger the angular error value, the larger the model error, and the lower the performance. The angular error is defined as
(1)$$L_{\mathrm{angular}\;}=arccos\left(\frac{g\cdot \hat{g}}{∥g∥\cdot ∥\hat{g}∥}\right)$$

Here, $g\in R^{3}$ is the true gaze vector, while $\hat{g}\in R^{3}$ is the predicted gaze vector of the model.

### 4.4. Window Size Parameter Analysis of Res-Swin-GE

Window size in Res-Swin-GE is an important parameter that determines the size of each sub-block of the feature map in W-MSA and SW-MSA. By adjusting the window size, the size of each sub-block can be controlled, thereby affecting the receptive field size, accuracy, and computational efficiency of the network. Reducing the window size can enhance the model’s ability to capture fine-grained details and extract local features. However, it is important to note that this also leads to increased computational requirements and memory usage. Whereas, increasing the window size can enhance the model’s ability to extract global features, capturing a broader context. However, it may result in the loss of detailed information and local features. This study keeps other parameters in Res-Swin-GE unchanged and discusses the influence of window size on the experimental performance of Res-Swin-GE. Table 1 lists the angle errors of Res-Swin-GE for different window sizes on the two public datasets. As listed in Table 1, this study evaluated the performance of window size from 1 to 7. Experiments show that windows that are too large or small degrade the performance of the model. This paper found that when adjusting window size = 2, Res-Swin-GE recorded optimal performance for MpiiFaceGaze and Eyediap, that is, Res-Swin-GE reached 3.75° on MpiiFaceGaze and 4.78° on Eyediap.

### 4.5. Experimental Results Analysis

#### 4.5.1. Angular Error of SwinT-GE and Res-Swin-GE on Different Subjects

As shown in Table 2, in the MpiiFaceGaze dataset, when the samples of P0 subjects were selected as the test set, the prediction performance of SwinT-GE and Res-Swin-GE was optimal, and the angular errors are 8.46° and 2.22°, respectively. When the samples from the P14 subjects were used as the test set, the prediction performances of SwinT-GE and Res-Swin-GE were the worst, with angular errors of 10.83° and 5.10°, respectively. The differences were 2.37° and 2.88°, respectively. As shown in Table 3, in the Eyediap dataset, when the samples of P14 subjects were selected as the test set, the prediction performance of SwinT-GE and Res-Swin-GE was optimal, and the angular errors are 8.28° and 3.18°, respectively. When the samples of P10 subjects were selected as the test set, SwinT-GE performed the worst, with an angular error of 10.75°. When the samples of P4 subjects were selected as the test set, Res-Swin-GE performed the worst with an angular error of 6.29°. The differences between SwinT-GE and Res-Swin-GE in the two cases were 2.47° and 3.11°, respectively.

As shown in Table 2 and Table 3, when the samples of different subjects in the dataset were selected as the training set, the performances of SwinT-GE and Res-Swin-GE differed significantly. This is because the training and test sets contain different subject samples. Even if they gaze in the same direction, different subjects show different kappa angles owing to the different internal structures of their eyeballs. The size of the kappa angle is determined by the internal parameters of the eyeball, and it is difficult to determine the size learned from the images. As a result, the performances of SwinT-GE and Res-Swin-GE were quite different when faced with different subjects. This is called the personal calibration problem, and is quite prevalent in the field of gaze estimation.

The calibration problem can be considered a domain adaptation problem, where the training set is the source domain and the test set is the target domain. The test set typically contains unseen subjects (cross-person questions) or unseen environments (cross-dataset questions) [40]. Personal calibration problems are related to many factors, such as the type of dataset. In the MpiiFaceGaze dataset, the difference between the extreme results predicted by SwinT-GE and Res-Swin-GE was 2.37° and 2.88°, respectively. In the EyeDiap dataset, the difference in extreme results predicted by SwinT-GE and Res-Swin-GE was 2.47° and 3.11°, respectively. This is because the MpiiFaceGaze dataset is collected from the real environment, has richer light changes and head pose changes, and contains more samples. The network structure proposed in this study can efficiently use the rich feature information contained in the dataset. Thus, the performance of this dataset is relatively stable.

#### 4.5.2. Angular Error of SwinT-GE and Res-Swin-GE at Different Gaze Angles

Figure 5a,b are the distribution diagram of gaze direction of subjects in the MpiiFaceGaze dataset and Eyediap dataset. The horizontal axis is the angle of the yaw axis of the gaze direction, and the vertical axis is the angle of the pitch axis of the gaze direction. The closer the color is to orange-red, the greater the data distribution, and the closer the color is to dark blue, the smaller the data distribution. Figure 5 illustrates that both datasets exhibit a higher concentration of gaze direction data in the central area, while displaying a lower distribution of data in the extreme gaze directions.

Figure 6a–d show the angular error distribution of the SwinT-GE and Res-Swin-GE prediction results on the two public datasets, respectively, where the horizontal axis is the angle of the yaw axis of the gaze direction and the vertical axis is the angle of the pitch axis of the gaze direction. The closer the color is to orange-red, the greater the angular error of the model’s prediction results at this gaze angle. the closer the color is to dark blue, the smaller the angular error of the model’s prediction results at this gaze angle. As shown in Figure 6, on the two datasets, SwinT-GE is overall bright, while Res-Swin-GE is overall blue. This shows that Res-Swin-GE performs better overall than SwinT-GE. However, the prediction performance of the two network structures under extreme gaze angles was poor because there were fewer learning samples under extreme angles, and the impact of personal calibration problems was more significant under extreme gaze angles.

Figure 7a,b show the improved prediction accuracy of Res-Swin-GE compared to SwinT-GE on the MpiiFaceGaze and Eyediap datasets under different gaze directions. Figure 7 was obtained by considering the difference between the angular error distribution of SwinT-GE and that of the dataset corresponding to Res-Swin-GE in Figure 6. The horizontal axis is the included angle of the yaw axis in the gaze direction, and the vertical axis is the angle of the pitch axis in the gaze direction. The closer the color is to orange-red, the greater the improvement in prediction performance at that gaze angle. the closer the color is to dark blue, the smaller the improvement. As shown in Figure 7, the color near the center is blue, while the color near the edges is orangish-red. This shows that the prediction accuracy of Res-Swin-GE showed a slight improvement in the gaze direction near the central area, but the prediction accuracy of Res-Swin-GE showed a more significant improvement in the extreme gaze direction.

#### 4.5.3. Comparison of Slicing-and-Mapping Mechanism and ResNet Block Feature Extraction Effect

In Figure 8 and Figure 9, column a shows the original input image, and columns b, c, d are extracted from the slicing-and-mapping mechanism, ResNet Block shallow convolution, and ResNet Block deep convolution feature maps. By comparing columns b, c in Figure 8 and Figure 9, it can be seen that the feature maps extracted by the slicing-and-mapping mechanism are sketchy. As a result, many of the fine-grained features of the eyes and coarse-grained features of the face are lost, which is very important for gaze estimation. In contrast, the feature map extracted by ResNet Block shallow convolution can contain more details and preserve various features of the image, which is helpful for further abstract learning of the subsequent structure.

Observing the column d feature maps in Figure 8 and Figure 9, it can be observed that the advantage of using ResNet Block to replace the slicing-and-mapping mechanism is not only that it effectively preserves various details of the image, but also that ResNet Block can automatically extract important features related to gaze estimation tasks (such as eye features) and filter out some weakly correlated features (such as nose, mouth, and other features), ensuring that the subsequent structure focuses on important features during the gaze prediction action tasks.

#### 4.5.4. Comparison to the State-of-the-Art

To better evaluate the performance of the Res-Swin-GE network, we compare Res-Swin-GE with state-of-the-art methods. The result is shown in Table 4.

Figure 10 shows the angular errors of Res-Swin-GE and the aforementioned networks, of which some recorded good performances on the MpiiFaceGaze and Eyediap datasets. The vertical axis is the angular error of each network on the MpiiFaceGaze, and the horizontal axis is the angular error of each network on Eyediap. the closer to the lower left corner, the better the performance of the method on the MpiiFaceGaze and Eyediap datasets. As shown in Table 4, the angular error of Res-Swin-GE on the MpiiFaceGaze dataset reaches 3.75°, and the angular error of the next best method (AFF-Net) on the MpiiFaceGaze dataset is 3.73°. the difference between the two is only 0.02°. Although Res-Swin-GE achieved the most advanced performance with an angular error of 4.78° on the Eyediap dataset, the error of the next best method (GazeTR-Hybrid) on the Eyediap dataset was 5.17°, and the difference between the two was 0.38°. Res-Swin-GE increased by 7.5% compared with GazeTR-Hybrid. It is evident that Res-Swin-GE outperforms other methods and is located closest to the bottom left corner of Figure 10.

### 4.6. Ablation Study

This section describes the experimental ablation results. Res-Swin-GE comprises ResNet18 and a Swin-T main structure. In order to verify the effectiveness of the combination of the two, this paper disassembles the two parts and adds a fully connected layer after the last layer of ResNet18 to output the gaze vector. In this paper, this structure is called ResNet18-Pure. The Swin-T part uses the SwinT-GE mentioned in Section 3.1, sets the classification number to two, and directly predicts the gaze vector from the facial image.

In Table 5, the angular errors of Res-Swin-GE on the MpiiFaceGaze and Eyediap datasets are 3.75° and 4.78°, respectively. After removing the Swin-T structure, the angular errors of ResNet18-Pure on the MpiiFaceGaze and Eyediap datasets are 4.00° and 5.15°, respectively. Compared with Res-Swin-GE, the angular errors increased by 6.7% and 7.7%, respectively, demonstrating the role of Swin-T structure in Res-Swin-GE. After removing ResNet18, the angular errors of SwinT-GE on the MpiiFaceGaze and Eyediap datasets reached 9.29° and 9.99°, respectively. Compared with Res-Swin-GE, the accuracy decreased significantly, and the angular error increased, i.e., 147.7% and 109.0%, respectively. This demonstrates the importance of the ResNet18 structure in Res-Swin-GE. Figure 11 presents a more intuitive comparison. It is apparent that the ResNet18 structure brings significant improvement. Meanwhile, the Swin-T architecture is also crucial.

## 5. Conclusions

In this paper, we introduced a Swin Transformer into the field of gaze estimation. We proposed two forms of Swin Transformers: the pure Swin Transformer (SwinT-GE) and the hybrid Swin Transformer (Res-Swin-GE). The structure of SwinT-GE followed the tiny version of the Swin Transformer, which directly predicted the direction of gaze from face images. Res-Swin-GE replaced the slicing-and-mapping mechanism of SwinT-GE with ResNet18. The experimental results demonstrated that Res-Swin-GE performed significantly better than SwinT-GE in gaze estimation tasks. Compared to state-of-the-art methods, Res-Swin-GE demonstrated strong competitiveness on both the Mapiifacegaze dataset and the Eyediap dataset. In future work, we will focus on extracting more robust features to address the challenges posed by individual calibration issues. Furthermore, we plan to apply Res-Swin-GE in the fields of virtual reality and augmented reality to provide more realistic and interactive experiences.

## Figures and Tables

**Figure 1 sensors-23-06226-f001:**
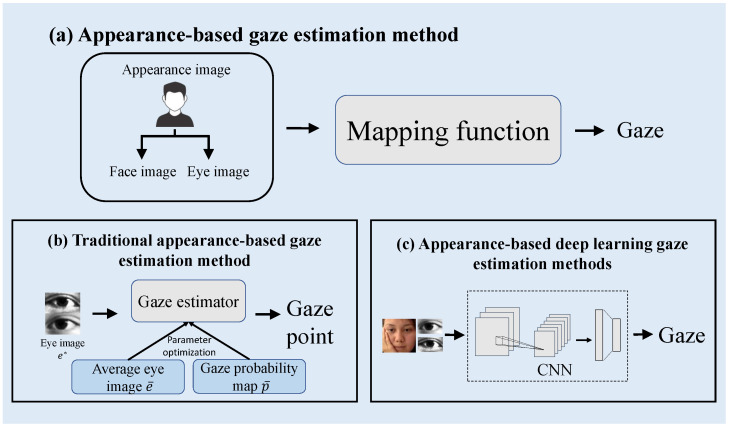
Basic methods for gaze estimation: (**a**) Appearance-based gaze estimation method. (**b**) Traditional appearance-based gaze estimation methods, where * represents a specific eye image. (**c**) Appearance-based deep learning gaze estimation methods.

**Figure 2 sensors-23-06226-f002:**
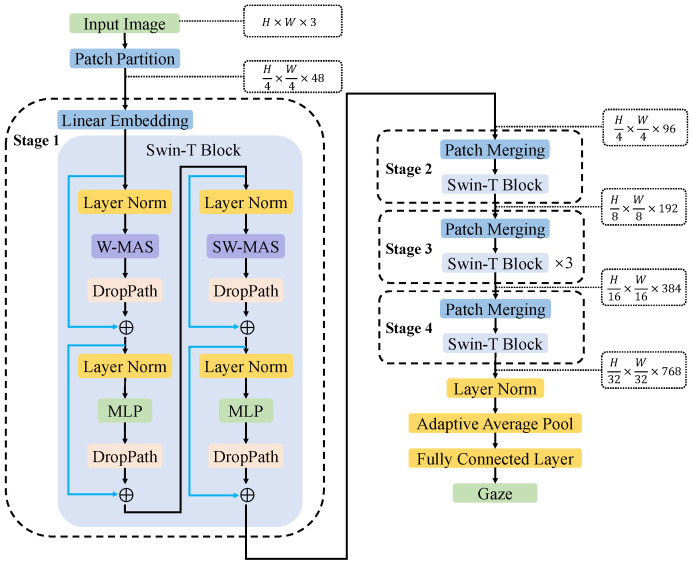
SwinT-GE network structure for gaze estimation.

**Figure 3 sensors-23-06226-f003:**
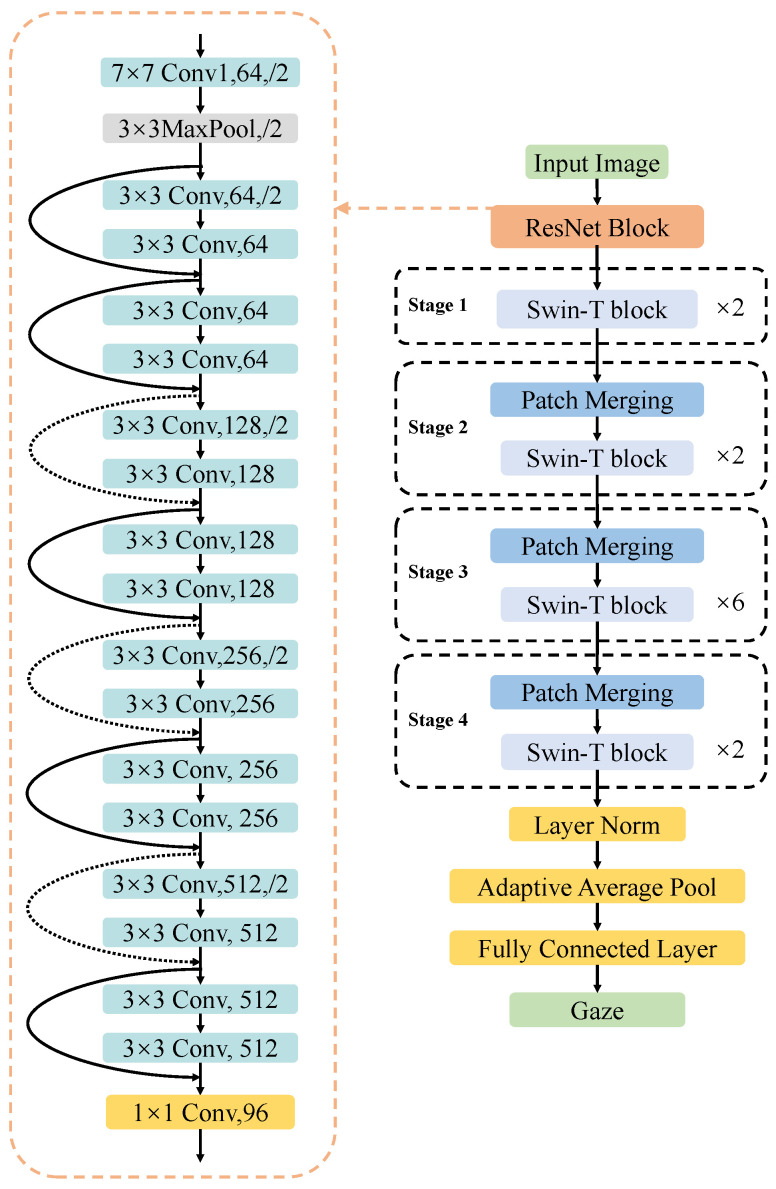
Network structure of Res-Swin-GE.

**Figure 4 sensors-23-06226-f004:**
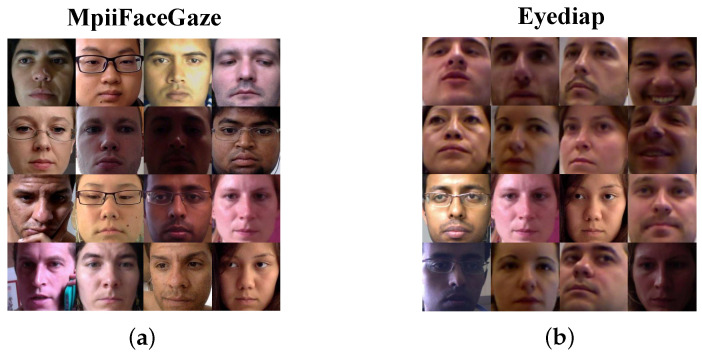
(**a**) Example face images in MpiiFaceGaze. (**b**) Example face images in Eyediap.

**Figure 5 sensors-23-06226-f005:**
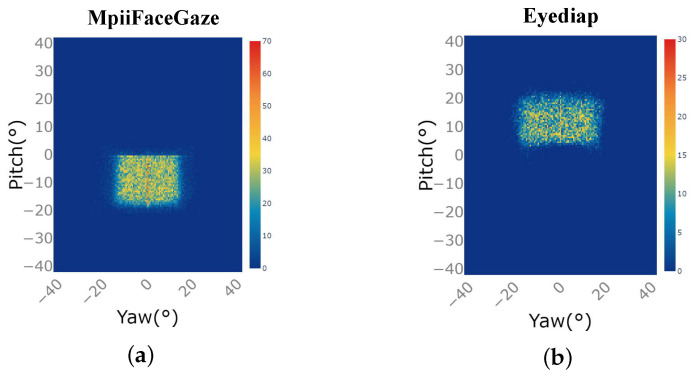
Data distribution of subjects’ gaze directions in the two public datasets: (**a**) MpiiFaceGaze [38]. (**b**) Eyediap [39].

**Figure 6 sensors-23-06226-f006:**
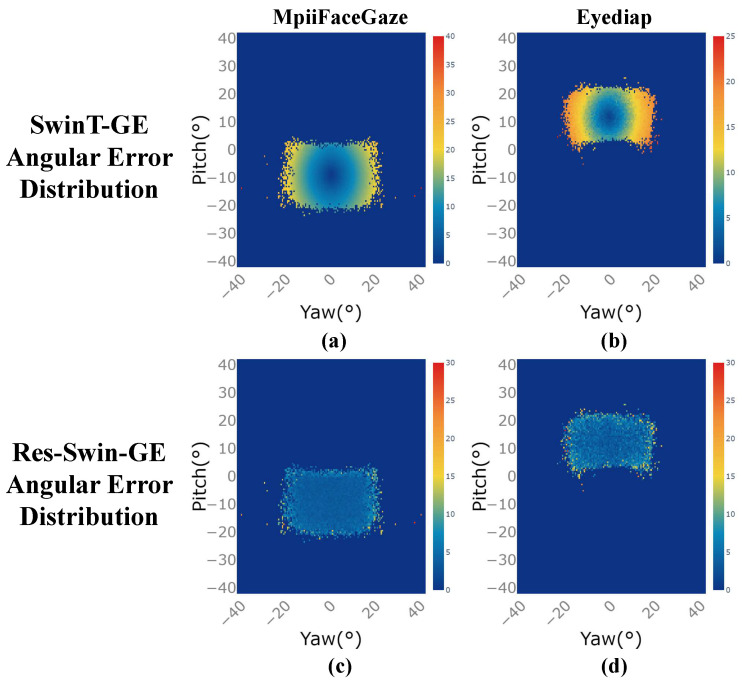
Angular error distribution of prediction results for the two network structures under different gaze directions on the two public datasets. (**a**,**c**) Angular error distribution of SwinT-GE and Res-Swin-GE on the MpiiFaceGaze [38] dataset. (**b**,**d**) Angular error distribution of SwinT-GE and Res-Swin-GE on the Eyediap [39] dataset.

**Figure 7 sensors-23-06226-f007:**
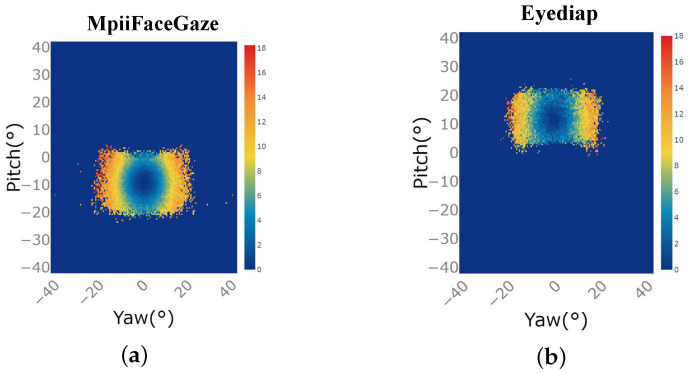
Improved prediction accuracy of Res-Swin-GE compared to SwinT-GE under different gaze directions for the two public datasets: (**a**) MpiiFaceGaze [38]. (**b**) Eyediap [39].

**Figure 8 sensors-23-06226-f008:**
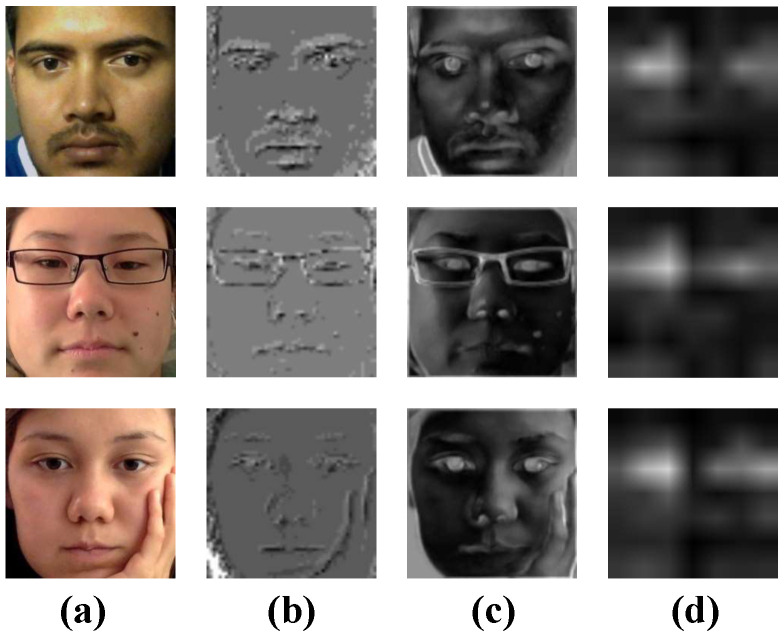
Example of feature maps extracted by different modules from images in the MpiiFaceGaze [38] dataset: (**a**) Original input image. (**b**) The feature maps extracted from the slicing-and-mapping. (**c**) The feature maps extracted from ResNet Block shallow convolution. (**d**) The feature maps extracted from ResNet Block deep convolution.

**Figure 9 sensors-23-06226-f009:**
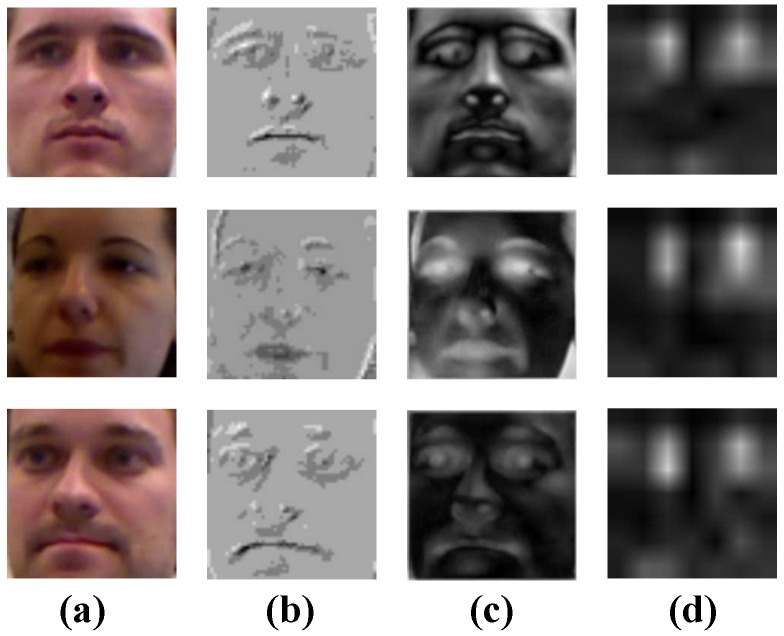
Example of feature maps extracted by different modules from images in the Eyediap [39] dataset: (**a**) Original input image. (**b**) The feature maps extracted from the slicing-and-mapping. (**c**) The feature maps extracted from ResNet Block shallow convolution. (**d**) The feature maps extracted from ResNet Block deep convolution.

**Figure 10 sensors-23-06226-f010:**
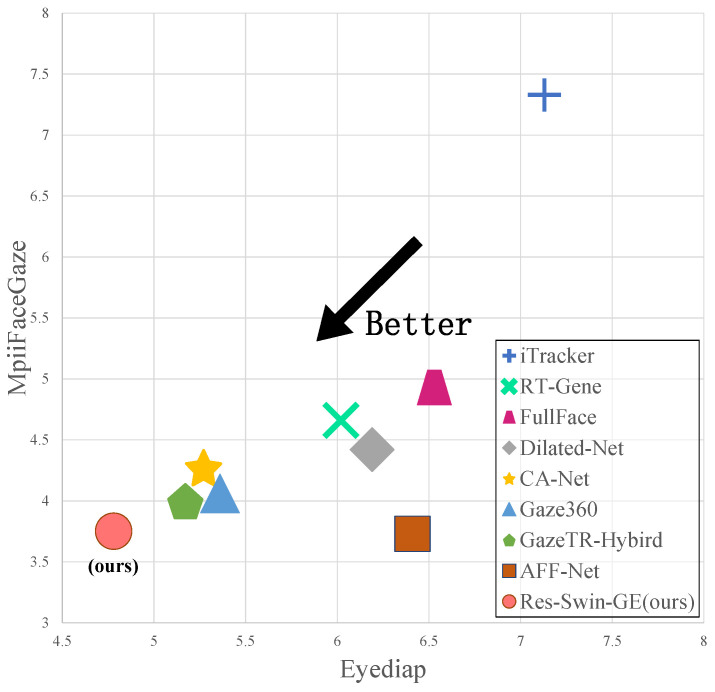
Angular errors of Res-Swin-GE compared to different networks on two public datasets.

**Figure 11 sensors-23-06226-f011:**
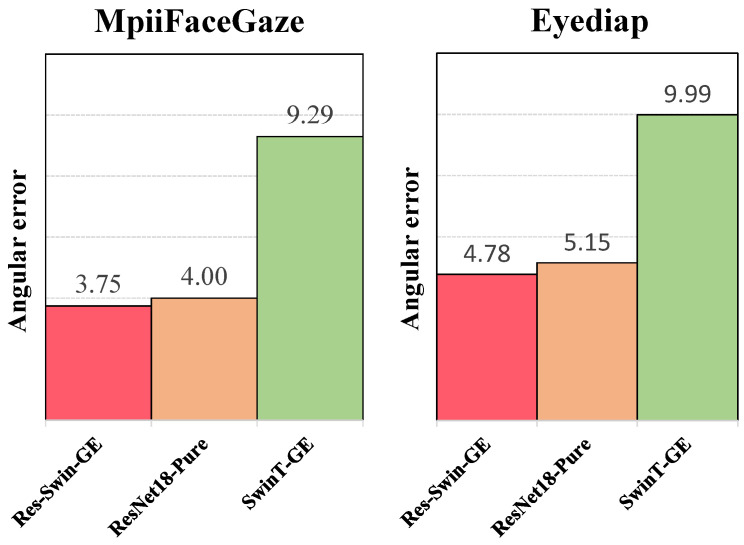
Angular errors of Res-Swin-GE, ResNet18-Pure, and SwinT-GE on the two publicly available datasets.

**Table 1 sensors-23-06226-t001:** Angular error of Res-Swin-GE on MpiiFaceGaze and Eyediap under different window sizes.

Hyper-Parameters		MpiiFaceGaze [38]	Eyediap [39]
Window Size WS	WS = 1	3.94°	4.86°
	WS = 2	3.75°	4.78°
	WS = 3	4.76°	4.85°
	WS = 4	3.82°	5.18°
	WS = 5	5.13°	4.98°
	WS = 6	4.67°	4.91°
	WS = 7	4.60°	4.95°

**Table 2 sensors-23-06226-t002:** Angular errors of SwinT-GE and Res-Swin-GE for each experimental subject on the MpiiFaceGaze [38] dataset.

Methods	P0	P1	P2	P3	P4	P5	P6	P7
SwinT-GE	8.46°	8.69°	8.89°	9.80°	9.08°	8.81°	10.32°	9.54°
Res-Swin-GE	2.22°	2.66°	3.60°	3.84°	3.01°	3.49°	3.16°	4.80°
Methods	P8	P9	P10	P11	P12	P13	P14	Avg
SwinT-GE	9.79°	9.08°	8.74°	8.51°	8.65°	9.11°	10.83°	9.29°
Res-Swin-GE	4.41°	4.48°	3.33°	3.49°	4.83°	3.79°	5.10°	3.75°

**Table 3 sensors-23-06226-t003:** Angular errors of SwinT-GE and Res-Swin-GE for each experimental subject on the Eyediap [39] dataset. Note that the Eyediap dataset lacks experimental subjects P12 and P13.

Methods	P1	P2	P3	P4	P5	P6	P7	P8	P9
SwinT-GE	10.36°	9.67°	9.84°	10.63°	10.48°	10.45°	10.68°	10.44°	9.21°
Res-Swin-GE	4.46°	4.15°	3.62°	6.29°	5.07°	6.21°	6.15°	5.82°	6.14°
Methods	P10	P11	P12	P13	P14	P15	P16	Avg	
SwinT-GE	10.75°	9.55°	-	-	8.28°	9.59°	9.95°	9.99°	
Res-Swin-GE	5.00°	4.16°	-	-	3.18°	3.39°	3.32°	4.78°	

**Table 4 sensors-23-06226-t004:** Comparison of angular errors of different networks on two public datasets.

Methods	MpiiFaceGaze [38]	Eyediap [39]
iTracker [10]	7.33°	7.13°
RT-Gene [41]	4.66°	6.02°
FullFace [42]	4.93°	6.53°
Dilated-Net [22]	4.42°	6.19°
CA-Net [12]	4.27°	5.27°
Gaze360 [23]	4.06°	5.36°
GazeTR-Hybrid [35]	4.00°	5.17°
AFF-Net [43]	3.73°	6.41°
Res-Swin-GE (ours)	3.75°	4.78°

**Table 5 sensors-23-06226-t005:** Angular errors of Quad Res-Swin-GE, ResNet18-Pure, and SwinT-GE on two public datasets.

Method	ResNet	Swin-T	MpiiFaceGaze [38]	Eyediap [39]
SwinT-GE	×	*√*	9.29°	9.99°
ResNet18-Pure	*√*	×	4.00°	5.15°
Res-Swin-GE	*√*	*√*	3.75°	4.78°

## Data Availability

Not applicable.

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
