# Peer review of "Gaze Estimation Based on Convolutional Structure and Sliding Window-Based Attention Mechanism"

_sensors, 2023, doi:10.3390/s23136226_

Round 1

Reviewer 1 Report

In this manuscript, the authors introduced a Swin Transformer and designed two forms into gaze estimation networks, which are SwinT-GE with pure Swin-T structure and Res-Swin-GE structure mixed with CNN and Swin-T, concerning to the study of human behavior in the level of attention and cognitive state. By showing angular error of two models on different subjects, the authors proved the advantage of ResNet which can enable the construction of deep convolutional neural networks that exhibit improved performance by analyzing the window size parameter of Res-Swin-GE and in ablation study. The impact on personal calibration problems was more significant under extreme gaze angles, which made model's performance poor, Also, made it clear that extracting more robust features to address the challenges posed by the personal calibration problem is vital. Therefore, I suggest it can be accepted after the author complements the content, articulates the obscure content and corrects the format.

1.     The specific structure of SwinT-GE network is a bit difficult to understand. For example, should it be briefly introduced on Page 5, Line 196, 199, what are these ‘’the feature map” refer to and could it be better to show the specific output of different layers.

2.     Figure notes and format issues are not uniform. For example, “Figure 1.”and’’Figure 11.’’ should be the same as other figures. Please double check the length of the space character.

3.     Suggested that nonoverlapping be changed to non-overlapping, subblock to sub-block, respectively on Page 3, Line 86, and Page 8, Line 289.

4.     On Page 9, Line 312, “Table 3” has no hyperlinks.

5.     Also on Page 9, Table 2, there is a problem with spacing of the table lines marked with “Methods”, Table 3 has the same problem.

6.     On Page 10, Table 3, where are the methods P12 and P13? And the conclusion about these two methods should be check twice on Page 9, Line 313 to Line 317.

7.     On Page 14, Line 413, ‘’are3.75°’’should be changed to ‘’are 3.75°’’

It could be better to use more active voice in your paragraph. For example, On Page 5, Line 199, the last sentence of paragraph, it may be easier to understand by revising as ''Adding the feature map from the second DropPath layer to the feature map can form a residual connection output of Swin-T Block before the feature map enters the second LN layer'' than before.

Reviewer 2 Report

The paper contains some new materials. However, some things need to be clarified.

Therefore, the authors should update the paper by incorporating my suggestions.
- What problem was studied, and why is it important?
- What methods were used?
- What are the significant results?
- What is the novelty of the work, and where does it go beyond previous efforts in the literature?
- What does the current paper add to the subject area compared with other published studies?
- The paper's results pave the way to new avenues that are fully awaited. Therefore, future works should be added in the conclusion part.
- Elaborate on the results depicted in all Figures.

- Update the review literature with recent relevant work from MDPI publisher.

- The English writing can be improved as some typos have been noticed in the manuscript; please correct all of them for the resubmission. 

Minor editing of English language required

Reviewer 3 Report

Dear Authors,

The topic is interesting but there are some rooms to improve:

1. Abstract is too long

2. Introduction is too long

3. Conclusion part is weak and maybe the most important part need to serious revision. Please revise your conclusion with more details and more information. I read this conclusion and it can not support your interesting literature and explanations in this paper.
